# In Silico and In Vitro Evaluation of the Antifungal Activity of a New Chromone Derivative against *Candida* spp.

**DOI:** 10.3390/biotech13020016

**Published:** 2024-05-25

**Authors:** Gleycyelly Rodrigues Araújo, Palloma Christine Queiroga Gomes da Costa, Paula Lima Nogueira, Danielle da Nóbrega Alves, Alana Rodrigues Ferreira, Pablo R. da Silva, Jéssica Cabral de Andrade, Natália F. de Sousa, Paulo Bruno Araujo Loureiro, Marianna Vieira Sobral, Damião P. Sousa, Marcus Tullius Scotti, Ricardo Dias de Castro, Luciana Scotti

**Affiliations:** 1Department of Clinical and Social Dentistry, Federal University of Paraíba, Campus I, João Pessoa 58051-900, PB, Brazil; gleyciaraujo0963@gmail.com; 2Postgraduate Program in Dentistry, Department of Clinic and Social Dentistry, Center of Health Sciences, Federal University of Paraíba, João Pessoa 58051-900, PB, Brazil; pallomachristine@gmail.com (P.C.Q.G.d.C.); paulalimanog@gmail.com (P.L.N.); dnobregaalves@msn.com (D.d.N.A.); pablo-rayff@hotmail.com (P.R.d.S.); 3Postgraduate Program in Natural and Synthetic Bioactive Products, Federal University of Paraíba, João Pessoa 58051-900, PB, Brazil; alanarodriguesferreira@hotmail.com (A.R.F.); jessicafarmacia2017@gmail.com (J.C.d.A.); nataliafsousa@ltf.ufpb.br (N.F.d.S.); paulobrunoaloureiro@ltf.ufpb.br (P.B.A.L.); mariannavbs@gmail.com (M.V.S.); damiao_desousa@yahoo.com.br (D.P.S.); mtscotti@gmail.com (M.T.S.); rcastro@ccs.ufpb.br (R.D.d.C.); 4Health Sciences Center, Federal University of Paraíba, Campus I, João Pessoa 58051-900, PB, Brazil; 5Institute of Drugs and Medicines Research, Federal University of Paraíba, Via Ipê Amarelo, S/N, João Pessoa 58051-900, PB, Brazil

**Keywords:** oral candidiasis, *Candida albicans*, antifungals, chromones

## Abstract

*Candida* species are frequently implicated in the development of both superficial and invasive fungal infections, which can impact vital organs. In the quest for novel strategies to combat fungal infections, there has been growing interest in exploring synthetic and semi-synthetic products, particularly chromone derivatives, renowned for their antimicrobial properties. In the analysis of the antifungal activity of the compound (*E*)-benzylidene-chroman-4-one against *Candida*, in silico and laboratory tests were performed to predict possible mechanisms of action pathways, and in vitro tests were performed to determine antifungal activity (MIC and MFC), to verify potential modes of action on the fungal cell membrane and wall, and to assess cytotoxicity in human keratinocytes. The tested compound exhibited predicted affinity for all fungal targets, with the highest predicted affinity observed for thymidylate synthase (−102.589 kJ/mol). MIC and CFM values ranged from 264.52 μM (62.5 μg/mL) to 4232.44 μM (1000 μg/mL). The antifungal effect likely occurs due to the action of the compound on the plasma membrane. Therefore, (E)-benzylidene-chroman-4-one showed fungicidal-like activity against *Candida* spp., possibly targeting the plasma membrane.

## 1. Introduction

The genus *Candida* comprises strains of pathogenic fungi commonly found on human reproductive and gastrointestinal mucosa, present in up to 80% of the healthy adult population [1]. Species within this genus are responsible for mucocutaneous infections (e.g., candidiasis) and invasive diseases that can affect vital organs [2]. These invasive infections increasingly cause death among humans, particularly in immunosuppressed individuals who are more susceptible to this type of infection [3].

*Candida* ranks as the third most prevalent cause of sepsis globally. Among the *Candida* species, *Candida albicans* consistently stands out as the predominant pathogen, responsible for 65.3% of *Candida* infections worldwide over the years. Latin American data show that *C. albicans* represents 51.8% of isolates, followed by other species such as *C. tropicalis*, *C. krusei*, *C. glabrata*, and *C. parapsilosis*. During the decade of the study, *C. albicans* was the most frequently isolated species, and a small percentage of these microorganisms isolated in Latin America were drug-resistant [4].

The onset of these infections is often attributed to an imbalance in the resident microbiota, leading to excessive growth of *Candida* spp. It can be triggered by local changes, exposure to immunosuppressive drugs, antibiotic use, and systemic diseases or immune alterations [1].

Oral candidiasis is a disease characterized by tissue invasion and excessive fungal growth of this opportunistic fungal disease of the oral mucosa [5]. Although *C. albicans* is the primary agent of oral candidiasis, other fungal strains are consistently associated with this infection, especially *C. glabrata* and *C. tropicalis* [5,6]. Thus, antifungal therapy is directly affected by the progressive association of these non-*albicans* species with fungal infections, which can result in the increasing resistance of these strains to available drugs [7,8].

There is a need to use effective treatments for proper patient management [9], considering that antifungal agents are restricted to a few compounds [10]. Primary agents include Nystatin, a polyene frequently used in clinical practice, and azole derivatives such as fluconazole [11]. Although considered first-line agents, limitations related to toxicity and the low susceptibility of non-*albicans* strains to these drugs suggest the merits of exploring new therapeutic approaches [2,3]. This is justified by the limited number of molecular targets to be explored in drug development due to the evolutionary proximity between humans and fungi, both eukaryotes [3]. Fungal resistance to at least one of these classes significantly limits the therapeutic options available and can lead to untreatable diseases [12].

As with antibiotics, excessive, inappropriate use of antifungals may lead to resistance. Fungi can become resistant to antifungals due to mechanisms such as genetic mutations, natural selection, selective pressure environments, and horizontal gene transfer. Antifungal resistance is a significant concern as it can impede the effective treatment of infections. Therefore, the prudent, responsible use of antifungal agents and the continued development of new drugs and strategies to combat resistant infections are important [13].

There is a need to identify new alternatives and therapeutic resources to combat these infections that have effective fungicidal activity while reducing resistance to new agents [14,15]. New synthetic and semisynthetic products have gained prominence, especially chromone derivatives, mainly due to their recognized antimicrobial activity [16,17].

Chromones are natural or synthetic compounds found mostly in plants and have anti-allergic, anti-inflammatory, antiviral, or antitumor activities [18]. Chromones and their derivatives are heterocycles and, with changes in their structures, they are useful in developing new therapeutic agents [19]. From chromone, it is possible to obtain 4-chromanone, a molecule of interest in medicinal chemistry whose derivatives are relevant for developing new drugs [20]. Figure 1 shows the structure of the compound (E)-benzylidene-chroman-4-one.

Due to the antifungal activity of this molecule (Figure 1), we carried out the reaction of 4-chromanone with benzaldehyde (used as an agent) in pyrrolidine to obtain the molecule (E)-benzylidene-chroman-4-one [21]. Thus, the present article aims to investigate the antifungal activity of (E)-benzylidene-chroman-4-one on different *Candida* species, its mechanism of action, and possible cytotoxic effects on human keratinocytes.

## 2. Materials and Methods

### 2.1. Procedure for Obtaining the AR25 Derivative

We added 0.084 mL (1.0123 mmol, 1.48 equiv.) of pyrrolidine to a mixture of 4-chromone (0.684 mmol) plus benzaldehyde (1.0123 mmol, 1.48 equiv.) dissolved in 2 mL of methanol (Figure 2). The solution was agitated at an ambient temperature for 24 h. Reaction was monitored by thin layer chromatography. Upon completion of the reaction (Figure 2), contents were transferred into cold water, resulting in precipitation. The precipitate was subsequently filtered, washed with water, dissolved in dichloromethane, and dried using anhydrous sodium sulfate (Na_2_SO_4_) [21].

#### Structure Analysis of Compound AR25 Is as Follows

AR25 is a white, crystalline solid and achieves a yield of 72% (0.492 mmol). 1H NMR (CDCl_3_, 400 MHz): δ 8.03 (dd; *J* = 7.9; 1.7 Hz; 1H); 7.88 (*s*; 1H); 7.51–7.40 (m; 4H); 7.31 (d; *J* = 6.8 Hz; 2H); 7.09–7.05 (m; 1H); 6.98–6.95 (m; 1H); 5.35 (d; *J* = 1.9 Hz; 2H). ^13^C RMN (100 MHz, CDCl_3_) δ 182.31; 161.24; 137.56; 135.96; 134.50; 131.03; 130.08; 129.56; 128.83; 128.05; 122.13; 122.01; 118.02; 67.71 [21].

### 2.2. In Silico Analysis

#### 2.2.1. Molecular Docking

The structures of AR25 and the reference drugs Nystatin and Miconazole were obtained from PubChem https://pubchem.ncbi.nlm.nih.gov (accessed on 30 October 2022) and imported into HyperChem™ software for energy minimization. This process utilized molecular mechanics (MM+) and semi-empirical (AM1) methods. The optimized structures were subsequently merged into a single SDF file using Standardizer 18.21.0 software, which also refined the structural data and converted the molecules into 3D. Fungal targets were sourced from the Protein Data Bank and included several enzymes: Exo-β-(1,3)-glucanase from *Candida albicans* (PDB: 1EQP, resolution 1.90 Å, X-ray diffraction method); AA11 lytic polysaccharide monooxygenase with Cu(I) from *Aspergillus oryzae* (PDB: 4MAI, resolution 1.40 Å, X-ray diffraction method); delta(14)-sterol reductase (PDB: 4QUV, resolution 2.74 Å, X-ray diffraction method); sterol 14-alpha demethylase (CYP51) from *Candida albicans* bound to the antifungal candidate VT1161 (PDB: 5TZ1, resolution 2.00 Å, X-ray diffraction method); and thymidylate kinase from *Candida albicans* (PDB: 5UIV, resolution 2.45 Å, X-ray diffraction method) [22].

The ligands and targets were subjected to molecular docking using Molegro Virtual Docker 6.0.1 (MVD), with IDs in Protein Data Bank (PDB) format. Standard settings were applied: MolDock Score for scoring; internal ES and internal Hbond for ligand evaluation; and validated Sp^2^-Sp^2^ torsions. The docking process involved ten runs using the MolDock SE algorithm (binding energy values in kJ/mol), with a maximum population size of 50 and a neighborhood distance factor of 1.00. The software was configured to generate up to 5 poses per ligand, with a maximum of 300 steps and 1500 interactions. A grid radius of 15 Å and a resolution of 0.30 were used to encompass the binding site of the macromolecules.

To validate the docking results, a redocking procedure was carried out to determine if the software accurately positioned the poses. This was assessed by calculating the root mean square deviation (RMSD), which measured the average deviation in distance between the enzyme and the superimposed standard ligand in its most stable pose. The same settings used for the analyzed substances were applied, with an acceptable RMSD value being less than 0.2 Å [23,24].

This metric was utilized to validate the crystal structure of sterol 14-alpha demethylase (CYP51) from *Candida albicans* in complex with the tetrazole-based antifungal drug candidate VT1161 (VT1) (PDB: 5TZ1). The active site coordinates were based on the co-crystallized ligand VT1161 (VT1). For the other enzymes under study, their active sites were determined using the site descriptions provided in the reference articles available in the PDB library. For targets lacking co-crystallized ligands, the coordinates were identified using the BiteNet—Skoltech I Molecule web servers for molecular pocket prediction (https://sites.skoltech.ru/imolecule/tools/bitenet (accessed on 3 November 2022)).

#### 2.2.2. Molecular Dynamics

Molecular dynamics (MD) simulations were performed using GROMACS 5.0 software, supported by funding from the Horizon 2020 Program of the European Union (Sweden), to evaluate the flexibility of protein-–ligand interactions [25,26]. The preparation of protein and ligand topologies was conducted using the GROMOS96 54a7 force field. Simulations were performed in a cubic box with the SPC water model for solvent, and system neutralization was achieved by adding Cl− and Na+ ions, followed by minimization to remove unfavorable contacts with solvent molecules. Temperature equilibration at 300 K utilized the V-rescale algorithm for 100 ps under NVT conditions, while pressure equilibration at 1 atm was accomplished using the Parrinello algorithm for up to 100 ps under NPT conditions. Molecular dynamics simulations encompassed 25,000,000 steps over 50 ns. Analysis included calculation of RMSD values for all Cα atoms relative to initial structures to assess structural flexibility and stability compared to experimental data. Additionally, RMSF values were examined to understand the roles of residues near the receptor binding site. Graphical representations were created using Grace software.

#### 2.2.3. ADMET Predictions

ADME parameters were predicted using the Swiss ADME open-access web tool (http://www.swissadme.ch). Toxicity predictions were carried out with the DataWarrior v. 4.7.2 software, known as OpenMolecules (http://www.openmolecules.org/datawarrior/download (accessed on 7 May 2024)) [27].

### 2.3. In Vitro Analysis

#### 2.3.1. Assessing the Minimum Inhibitory Concentration (MIC) and Minimum Fungicidal Concentration (MFC)

The MIC was determined following the microdilution protocol outlined in the Clinical and Laboratory Standards Institute (CLSI) guidelines from 2008 (CLSI, 2008). This assay was performed in triplicate by using reference strains of *Candida* species obtained from the American Type Culture Collection (ATCC, Rockville, MD, EUA): *C. albicans* ATCC 10231, *C. krusei* ATCC 6258, *C. glabrata* ATCC 90030, *C. tropicalis* ATCC 750, *C. parapsilosis* ATCC 22019, *C. albicans* ATCC 60193, and *C. albicans* ATCC 90028. Yeast suspensions were prepared in Roswell Park Memorial Institute Medium 1640 (RPMI) and adjusted to an optical density equivalent to 2.5 × 10^3^ colony-forming units per milliliter (CFU/mL) at 530 nm (absorbance 0.08–0.13). Sterile 96-well U-bottom microplates containing RPMI were used for microdilution. Initially, 100 μL of RPMI was added to each well in the plates. Subsequently, 100 μL of the test substance was added to the first well of each column, followed by serial dilution. Finally, 100 μL of the fungal inoculum was added to each well. The plates were then incubated for 24 h at 35 °C. Results were determined by observing fungal growth or cell aggregates at the bottom of the wells. The lowest concentration of the substance that visibly inhibited fungal growth was considered the MIC. The concentrations of the tested molecule ranged from 1000 to 7.81 µg/mL. Nystatin (Sigma-Aldrich, São Paulo, SP, Brazil) served as a positive control at an initial concentration of 48 µg/mL. Furthermore, viability assessments of the strains and sterility checks on the culture medium were conducted.

The CFM is defined as the lowest concentration capable of inhibiting fungal strain growth on solid medium. Aliquots of 20 μL corresponding to the MIC and two higher concentrations (2× MIC and 4× MIC) were plated on Sabouraud dextrose agar (KASVI, Kasv Imp, and Dist de Prod/Laboratorios LTDA, Curitiba, Brazil). The plates were then incubated at 35 °C for 24 h. Fungal growth in the culture medium was assessed visually to determine the results. The CFM/MIC ratio was calculated to assess whether the compound exhibited fungistatic activity (CFM/MIC ≥ 4) or fungicidal activity (CFM/MIC < 4) [28,29].

#### 2.3.2. Mode of Action 

##### Ergosterol Assay 

Due to the activity of certain antifungal agents on fungal plasma membrane ergosterol, either forming complexes or inhibiting membrane biosynthesis, an increase in MIC in the presence of exogenous ergosterol indicates that the studied molecule is impacting the fungal plasma membrane [30]. For this experiment, the microdilution method was employed, where 100 μL of culture medium was dispensed into each well. Next, 100 μL of the substances under investigation was added to the initial well of each column, followed by serial dilution, and then 100 μL of the fungal inoculum was introduced into each well. Exogenous ergosterol (Sigma-Aldrich, São Paulo, Brazil) was added to the microdilution plates at a concentration of 400 µg/mL. The plates were subsequently incubated at 35 °C for 24 h. Nystatin was included as a positive control. The results were assessed by observing cell aggregates at the bottom of the wells.

##### Sorbitol Assay (Cell-Wall-Related Effects)

Sorbitol functions as an osmotic protector that affects the fungal cell wall [30]. We employed the serial microdilution method, where 100 μL of culture medium was dispensed into each well of the plates. Following this, 100 μL of (E)-benzylidene-chroman-4-one solution was added to the initial well of each column, followed by serial dilution, and then 100 μL of the fungal inoculum was introduced into each well. Next, 0.8 M sorbitol (D-sorbitol anhydrous) (INLAB, São Paulo, Brazil) was added to the culture medium, and the plates were subsequently incubated at 35 °C for 24 h. The results were assessed by observing cell aggregates at the bottom of the wells. Caspofungin at 4 µg/mL served as a positive control [31].

#### 2.3.3. MTT Cell Viability Assay

The MTT cell viability assay was performed to evaluate the compound’s cytotoxicity. For this purpose, HaCaT cell line (human keratinocytes) cells were used, which are the main cells composing the epithelium of the oral mucosa [32]. The cells were cultured in DMEM medium containing 10% fetal bovine serum, 100 μg/mL of streptomycin, and 100 U/mL of penicillin, maintained at 37 °C in a humidified atmosphere with 5% CO_2_. For the assay, the cells were seeded into 96-well plates at a density of 3 × 10^5^ cells/mL. After 24 h, the cells were treated with the compound (6.25–400 μM) dissolved in Dimethyl Sulfoxide (DMSO) for 72 h. Doxorubicin (0.31–20 μM) served as a positive control. Subsequently, the supernatant was aspirated, and a solution of 3-(4,5-dimethylthiazol-2-yl)-2,5-diphenyltetrazolium bromide (MTT) (5 mg/mL) was added. Optical densities were then measured using a microplate reader (Synergy HT, BioTek-USA). Cell viability results are presented as percentages (mean ± SD) and were analyzed using one-way ANOVA followed by Tukey’s post-test (*p* ≤ 0.001). Additionally, the half-maximal inhibitory concentration (IC_50_) was calculated using nonlinear regression with 95% confidence intervals.

## 3. Results

### 3.1. In Silico Analysis

#### 3.1.1. Molecular Docking

Pharmacodynamic prediction analysis was conducted to assess the structural groups within the target compound that contribute to its antifungal activity. Prior to docking, redocking was simulated, calculated only for the target crystal structure of sterol 14-alpha demethylase (CYP51) from *Candida albicans* in complex with the tetrazole-based antifungal drug candidate VT1161 (VT1) (PDB: 5TZ1), which is the only 3D structure with a co-crystallized ligand. The value obtained was 0.6195, indicating that it is within the acceptable range for the root mean square deviation of the structure.

According to Table 1, the compound (E)-3-benzylidene-chroman-4-one showed negative binding energy values for the molecular targets, indicating a predicted affinity with all the targets evaluated. This was especially true for the thymidylate synthase target (PDB: 5UIV) (−102.589 KJ.mol^−1^), for which the test compound showed greater affinity compared to the control drug Nystatin (−91.3965 KJ.mol^−1^). The control drug Miconazole showed a score of −133.473 KJ.mol^−1^.

Figure 3 shows the analyzed docking maps of (E)-benzylidene-chroman-4-one with the thymidylate synthase target (PDB: 5UIV), in which the presence of hydrogen bonds can be detected (green dashed line). Hydrophobic interactions of the alkyl, pi–pi T-shaped, pi–alkyl, and pi–pi stacked type were also observed (pink dashed line). Figure 3A shows that the interaction between 5UVI (thymidylate synthase), (E)-3-benzylidene-chroman-4-one, Miconazole, and Nystatin occur through hydrogen bonding (shown in green), with amino acid residues Pro 37 (1 interaction) and Arg 92 (1 interaction) observed in the carbonyl function of the benzophenone group. In the benzene rings of the chemical structure of the compound, alkyl and pi–alkyl interactions were observed, established by residues Pro 37 (1 interaction), Phe 67 (1 interaction), and Tyr 161 (1 interaction).

Miconazole (control drug) exhibited similar hydrophobic interactions to the test compound (E)-benzylidene-chroman-4-one, including alkyl, pi–alkyl, and pi–pi stacked interactions with residues Pro 37 (1 interaction), Leu 51 (2 interactions), Arg 160 (1 interaction), Tyr 100 (1 interaction), Phe 67 (2 interactions), and His 64 (1 interaction). Hydrogen bonding interactions were also observed with Tyr 161 (1 interaction) and Glu 159 (1 interaction). Furthermore, electrostatic interactions such as pi–cation and pi–anion (dashed orange lines) occurred with amino acids Arg 39 (1 interaction), Arg 92 (1 interaction), and Glu 159 (1 interaction). Similar residues were observed between Miconazole and the test compound, involving residues Pro 37 and Phe 67 in alkyl and pi–alkyl interactions.

In contrast to (E)-benzylidene-chroman-4-one and azole derivative Miconazole, Nystatin, a polyene, showed a more polar character in the interactions identified, as a significant proportion of the interactions observed were hydrogen bonding interactions. These interactions involved residues Phe 67 (one interaction), Gly 97 (one interaction), Arg 92 (one interaction), Tyr 100 (one interaction), Glu 162 (one interaction), Glu 159 (two interactions), and Arg 39 (one interaction). In addition, hydrophobic interactions of the alkyl and pi–alkyl type were observed through residue Arg 92 (one interaction), and covalent interactions were observed through residue Pro 37 (one interaction). Another type of interaction observed was an unfavorable interaction (dashed red line) involving Tyr 161 (four interactions) and Asp 13 (one interaction). Similar residues were found between (E)-benzylidene-chroman-4-one and Nystatin, including Arg 92 for hydrogen bonding interactions and Pro 37 for hydrophobic interactions.

#### 3.1.2. Molecular Dynamics

Following the assessment of the potential antifungal activity of the test compound AR25 against key mechanisms, molecular dynamics simulations were conducted to investigate the enzyme’s flexibility and the stability of interactions in the context of various environmental factors such as solvent, ions, pressure, and temperature. This analysis serves to complement the docking results and determine whether the compound maintains strong binding to the enzyme in the presence of physiological factors found within the host organism. Thymidylate Kinase from *Candida albicans* (PDB: 5UIV) was selected for analysis due to AR25’s higher affinity for this protein. Subsequently, RMSD values were calculated for the Cα atoms of the enzyme–ligand complex and the individual ligand structures to assess their stability over time.

Concerning the Thymidylate Kinase target (PDB: 5UIV), the analysis of protein RMSD metrics (Figure 4) revealed that the complex involving compound AR25 (represented by the red line) exhibited superior stability compared to all other complexes studied, maintaining RMSD values ranging from 0.25 nm to 0.3 nm and up to the 20 ns mark. Notably, AR25 sustained these RMSD values thereafter, demonstrating sustained stability in comparison to the control compound Miconazole (illustrated by the green line) and Nystatin (depicted by the blue line), both of which displayed higher RMSD values, peaking at 0.33 nm until the 30 ns interval. Furthermore, the complex involving Thymidylate synthase protein (PDB: 5UIV), represented by the black line, exhibited the highest stability, with RMSD values at 0.25 nm. This stability of the Thymidylate Kinase protein from *Candida albicans* (PDB: 5UIV) is crucial for maintaining the binding of compounds to the active site.

Examining ligand stability in the presence of solvents (as illustrated in Figure 5), it was observed that the test compound AR25 (depicted by the red line) exhibited lower RMSD values compared to the control drugs Miconazole (in green) and Nystatin (in blue). Nystatin and Miconazole displayed significantly higher instability, evident from their elevated RMSD values. Consequently, in the presence of solvents, ions, and other factors, AR25 demonstrated the ability to form robust bonds with the active site. Thus, it is suggested that AR25 molecules are inclined to persist within the active site even amidst varying environmental conditions such as temperature, pressure, solvent, and ions.

To comprehend the flexibility of the residues and amino acids pivotal for conformational changes in the *Candida albicans* enzyme Thymidylate kinase (PDB: 5UIV), we computed the root mean square fluctuations (RMSF) for each amino acid within the protein. Amino acids with higher RMSF values indicate greater flexibility, while those with lower RMSF values signify reduced flexibility. Notably, amino acids exhibiting fluctuations exceeding 0.3 nm are deemed influential in channel structure flexibility. Our analysis revealed that, within the protein complexed with the AR25 compound (depicted in Figure 6), residues positioned at one significantly contribute to conformational alterations. Intriguingly, this residue does not constitute part of the protein’s active site. However, its involvement suggests a mechanism by which the AR25 compound remains bound within the active site.

Additionally, the evolution of protein packaging levels was also analyzed alone and in complex with the compound AR25 (red) and the controls Miconazole (green) and Nystatin (blue), using the radius of gyration (Rg) values (Figure 7). In summary, Rg provides an idea of how compact or extended a molecule is during a molecular dynamics simulation, providing information about the conformation, stability, interactions, and flexibility of the molecules under analysis. It was observed that the molecules AR25 (red) and Miconazole (Green), when coupled to proteins, do not present variations in relation to the protein alone (black), this indicates that these complexes are not likely to show fluctuations in the tertiary structure of the enzyme.

The Coulomb and Lennard-Jones interaction energies (Table 2) of the protein–ligand were determined to provide insights into the stability of interactions within the active site. Based on Coulomb energy calculations (C), the test compound AR25 demonstrated greater interaction stability with the active site of the enzyme when compared to the Miconazole control, thus demonstrating a large contribution of electrostatic interactions and hydrogen bonds. However, according to Coulomb metrics, the controls Miconazole and Nystatin presented greater stability, demonstrating a greater contribution of electrostatic energy in the established interactions.

#### 3.1.3. ADMET Predictions

Initially, predictions for absorption, distribution, metabolism, excretion, and toxicity (ADMET) were performed utilizing the Swiss ADME and Osiris DataWarrior platforms. The objective of this assessment was to pinpoint compounds exhibiting advantageous pharmacokinetic, pharmacodynamic, and pharmacological attributes, signifying their viability for subsequent advancement. Table 3 furnishes data concerning gastrointestinal absorption, blood–brain barrier permeability, mutagenicity, tumorigenicity, reproductive effects, and skin irritancy pertinent to the compound under investigation.

The potential interactions of the compound were investigated regarding the five major isoforms of cytochrome P450 (CYP1A2, CYP2C19, CYP2C9, CYP2D6, CYP3A4) (Table 4). Interestingly, only the compound bulbocapnine was predicted to have the potential to inhibit all CYP isoforms, especially the CYP2C19 and CYP2C9 isoforms, for which no other compound was predicted as an inhibitor.

The compound AR25’s gastrointestinal absorption and brain penetration were depicted through the Boiled Egg diagram, showcased in Figure 8, utilizing the Swiss ADME web tool. In this diagram, the white area denotes the region with the highest probability of absorption by the human gastrointestinal system, while the yellow portion (the yolk) signifies the area with the highest likelihood of brain penetration. Examination of the diagram revealed that all evaluated compounds reside within the yellow region, indicating a substantial likelihood of absorption by the human gastrointestinal tract and permeation into the brain.

### 3.2. In Vitro Analysis

#### 3.2.1. Determination of MIC and MFC

Table 5 presents the MIC and MFC values for (E)-benzylidene-chroman-4-one and Nystatin. The MIC and MFC values for the tested compound varied between 264.52 µM (62.5 µg/mL) and 4232.44 µM (1000 µg/mL). In the case of Nystatin, the MIC and MFC values ranged from 1.5 µg/mL to 3 µg/mL. Moderate antifungal activity was observed against most of the tested strains [33]. Regarding the MFC/MIC ratio, (E)-benzylidene-chroman-4-one has fungicidal activity against all strains tested.

#### 3.2.2. Mechanism of Action on the Fungal Membrane and Cell Wall

##### Ergosterol Assay (Cell-Membrane-Related Effects)

(E)-3-benzylidene-chroman-4-one was tested to evaluate its activity on the cell membrane of *Candida* strains. Table 6 shows an elevation in MIC when exogenous ergosterol is present (500 µg/mL to 1000 µg/mL in *C. albicans* and 62.5 µg/mL to 1000 µg/mL in *C. krusei*). This discovery implies a potential mechanism of action of the compound under investigation involving fungal plasma membrane function.

##### Sorbitol Assay (Cell-Wall-Related Effects)

(E)-3-benzylidene-chroman-4-one was subjected to tests to evaluate its activity on the cell wall of *Candida* strains. Table 7 displays that the MIC of the substance remained unchanged when compared to MIC without the addition of sorbitol, against the two evaluated strains. The MIC values do not change in the presence of sorbitol. This result indicates that the compound does not act by disrupting the fungal cell wall. In contrast, the control drug (Caspofungin) exhibited an increase in MIC in the presence of sorbitol.

#### 3.2.3. MTT Cell Viability Assay

(E)-3-benzylidene-chroman-4-one significantly reduced the viability of HaCaT cells only from a concentration of 50 μM (Figure 9). From 6.25 to 25 µM, cell viability remained unaffected significantly by the compound, with reductions not exceeding 20%. The half-maximal inhibitory concentration (IC_50_) value for the compound was 36.25 μM, suggesting low cytotoxicity. On the other hand, doxorubicin, a standard cytotoxic drug, induced high cytotoxicity (IC_50_ = 0.28 μM). The data show a relative margin of non-toxic concentrations, as observed in Figure 9. For a better definition of the cytotoxic potential. It is recommended to assess additional parameters to establish its safety for use in different cell types and in vivo models.

## 4. Discussion

Molecular docking is a key technique in structural molecular biology used to predict how small molecules bind to specific receptors by estimating their binding modes and affinities experimentally. It is widely used as a standard computational approach to optimize key compounds and for virtual screening studies in the search for new molecules with biological activity [34]. Simply put, molecular docking analyzes how two structures, usually a protein called the receptor and another called the ligand, interact with each other [35].

In the present study, the binding values of (E)-benzylidene-chroman-4-one and two drugs used to treat oral candidiasis (Nystatin and Miconazole) were analyzed using molecular docking [36]. Analyzing the docking of these molecules allows for the calculation of the energy values necessary to allow coupling to the possible binding sites [37]. The lower the energy value, the stronger the binding site [38].

Thus, (E)-3-benzylidene-chroman-4-one demonstrated affinity with all of the evaluated targets, showing negative values for the binding energies. Docking maps of (E)-3-benzylidene-chroman-4-one with the analyzed targets indicated the presence of hydrogen bonds, which are stronger bonds that result from the electrostatic attraction between a negatively charged atom and a hydrogen atom; steric interactions, which are internal bonds; and electrostatic interactions, resulting from the electrostatic attraction between ionized groups with opposite charges, characterized as weak bonds [39]. The target Thymidylate kinase (5UIV), a phosphotransferase crucial for synthesizing genetic material and parasite survival [40], exhibited the highest affinity with the test compound.

The values obtained for the MIC of the compound ranged from 62.5 to 1000 µg/mL and are consistent with the results obtained for the MFC. The MFC represents the minimum concentration required to inhibit fungal strain growth on solid media, providing a means to evaluate the antifungal activity of new compounds and validate findings obtained from MIC testing [41,42].

Stanley, Ifeanyi, and Eziokwu (2014) used microdilution and MIC rates for an aqueous extract of *Aloe vera* gel, a plant species that contains chromones in its chemical composition. For *C. albicans*, the MIC was 0.5 mg/mL. Similar results were obtained for the ethanolic extract. When these results were compared with those obtained in this study, it was observed that the MIC against *C. albicans* was even lower for the analog tested.

According to the classification that standardizes the antifungal potency of compounds screened against *Candida* using interpretative cutoff points for MIC values of bioactive synthetic and semisynthetic compounds, (E)-benzylidene-chroman-4-one was found to have moderate antifungal activity [33]. This bioactivity classification is proposed for compounds with MIC values ranging between 26 and 100 μg/mL [33].

In a study by Iwashima et al. [43], *Candida* species showed good sensitivity to 13-hydroxyphyllosporin, a compound isolated from marine algae, which contains the metabolite chromones in its constitution. This compound inhibited the growth of four out of four strains (100%) and achieved an MIC of 256 μg/mL. The control with the standard antifungal, in this case, Nystatin (polyene class drug) at 100 IU/mL, inhibited the growth of three out of four (75%) *Candida* strains.

Metal complexes with chromone-derived ligands also show a wide range of bioactivities. In a study by Tamayo et al. [44], the antifungal activity of hydrazones was evaluated in combination with silver chromone derivatives. This combination showed antifungal activity against all *Candida* strains and was more potent than Nystatin (polyene class drug).

A study by Malefo et al. [45] describes the synthesis of two chromone derivatives (oxepinochromones) and the effect against *Cryptococcus neoformans* and *Candida albicans*. Significant activity was observed against both fungi. The highest activity against *C. neoformans* was displayed by 12-O-acetylptaeroxylol, with a MIC value of 4.9 μM. In contrast, 12-O Acetylerianthrin demonstrated the highest activity against *C. albicans*, with a MIC of 9.9 μM.

In a study by Chohan et al. [46], a series of sulfonamides (sulfanilamide, sulfaguanidine, sulfamethoxazole, 4-aminoethyl-benzenesulfonamide, and 4-amino-6-trifluoromethylbenzene-1,3-disulfonamide) derived from chromones, previously reported as carbonic anhydrase inhibitors, were tested for antibacterial and antifungal activity (*Trichophyton longifusus*, *Candida albicans*, *Aspergillus flavus*, *Microsporum canis*, *Fusarium solani*, *Candida glabrata*). Regarding antifungal activity, only the derivative 4-aminoethyl-benzenesulfonamide showed no activity.

Prakash, Kumar, and Parkash [47] investigated the antifungal activity of seven chromones against phytopathogenic fungi. In vitro testing of these seven compounds against three strains demonstrated superior antifungal activity compared to cycloheximide.

Chromones are synthetic or natural compounds found especially in plants and perform various biological activities [18]. Through the reduction of chromone, 4-chromanone can be obtained. The compound (E)-benzylidene-chroman-4-one was synthesized through a reaction between 4-chromanone and benzaldehyde, in the presence of pyrrolidine [21]. Chromones have various pharmacological actions documented in the literature, such as potent anti-inflammatory action, making them useful in treating allergic rhinitis and asthma. They are used in drugs (e.g., cromolyn and nedocromil) as active ingredients [15]. Chromones also exhibit antitumor [48] and antioxidant [49] activities, among others. In addition to these mentioned actions, chromones have previously described antifungal properties [17]. Following this perspective and considering the results obtained by Ferreira et al. [21], through preliminary studies in which (E)-benzylidene-chroman-4-one was previously synthesized and characterized, the compound was defined as a potential option for the development of new drug with high antifungal activity for the treatment of candidiasis. Thus, tests were conducted to elucidate possible mechanisms of the molecule’s antifungal response.

The ongoing development of new antifungals is crucial to address the growing challenges of antifungal resistance and ensure that patients have access to effective and safe treatments for fungal infections. This requires investment in the research into and development of new compounds, as well as the implementation of strategies to promote rational use of existing antifungals. While *C. albicans* is commonly implicated in invasive infections in many healthcare settings, non-albicans species have globally emerged with increased rates of antifungal resistance. This resistance is particularly notable against fluconazole, which is relatively affordable and well-tolerated, and, consequently, has been extensively utilized for treating these infections [13].

Overall, drugs currently used to treat fungal infections caused by *Candida* yeasts target fungal cell structures, particularly components of the cell wall and plasma membrane [14,33,50]. However, due to drug resistance developed by strains, the search for new therapeutic alternatives has increased significantly. The mechanism of action of antifungals can operate through an interaction between the compound and components of the fungal membrane or cell wall. When the compound targets the plasma membrane, its action can cause instability in the fungal cell membrane, particularly through the direct binding of the antifungal agent to ergosterol present in the pathogen, resulting in damage to the cell membrane and leading to fungal death. However, it can also act by inhibiting ergosterol synthesis, consequently modifying the permeability of the cell membrane [50,51]. The current study investigated the potential mechanism of action of the compound (E)-benzylidene-chroman-4-one using methodologies that consider possible antifungal effects resulting from alterations in functions involving the fungal membrane and/or cell wall.

We conducted tests with culture media plus ergosterol or sorbitol to determine the action on fungal cell walls/membranes, respectively. The test molecule, when in the presence of exogenous ergosterol, showed an increase in MIC in both strains evaluated. A similar result was observed for Nystatin (polyene class drug), which binds to steroids in the fungal cell membrane, causing changes in membrane permeability and consequently influencing the leakage of cytoplasmic contents [4,50]. However, in the presence of sorbitol, the tested compound did not show a change in MIC, differing from Caspofungin, which exhibited an increase in the minimum inhibitory concentration.

Ergosterol is the predominant sterol in fungi and a fundamental constituent of the cell membrane, playing a critical role in membrane integrity, function, and biosynthesis [52,53]. Consequently, an increase in the MIC in the presence of exogenous ergosterol in the culture medium suggests that the tested compound could potentially act through mechanisms involving the cell membrane [53,54]. These results confirm the findings obtained in the molecular docking study, since the tested compound showed affinities for squalene epoxidase, δ-14-sterol reductase, and 14-α-demethylase, enzymes recognized for their role in ergosterol synthesis [55].

Sorbitol serves as an osmotic protector and can be used to assess whether a molecule acts on the fungal cell wall or not [56]. Thus, the absence of changes in the MIC of the tested compound, in the presence of sorbitol in the culture medium, indicates that its action is not related to the fungal cell wall [57,58,59]. In this study, it is observed that (E)-benzylidene-chroman-4-one does not have a mechanism of action defined by interference with structures involved in the functions of the cell wall, although the in silico test predicted an affinity of the compound with the enzyme 1,3β-glucan synthase. This can be explained by the lack of energetic stability of the predicted complex, as well as the conformational stability of the compound. To further assess this hypothesis, monitoring the stability by calculating the root mean square deviation (RMSD) is recommended [11].

Compared to analogous molecules such as coumarin-derived compounds, which are heterocyclic compounds of natural or synthetic origin [21], (E)-benzylidene-chroman-4-one showed similar results for cytotoxic effects on human keratinocytes. The coumarin derivatives showed moderate cytotoxicity at 100 μM) [60]. Other studies have also reported low cytotoxicity of coumarin derivatives, with an IC50 expression of 55.5 μM on human keratinocyte [61], suggesting the need for further cytotoxic analyses with (E)-benzylidene-chroman-4-one as the results are promising compared to similar compounds. Therefore, conducting in vivo studies is advised to assess additional parameters of toxicity.

The in vitro and in silico tests employed in this study provide strong control of confounding variables. However, the next stages of investigation should include methodological proposals that consider in vivo and clinical models, enabling a better understanding of the antifungal effect as well as assessment of the safety and efficacy of the product.

## 5. Conclusions

(E)-3-benzylidene-chroman-4-one showed the predicted affinity for all fungal targets according to the in silico model, exhibiting interaction with the targets equal to or superior to the controls. Additionally, the compound demonstrated potential fungicidal activity against strains of *Candida* species (ATCC), possibly involving interactions with enzymes involved in ergosterol synthesis, an essential component of the fungal plasma membrane. Therefore, these pre-elimination assays make AR25 a pharmacologically interesting compound as a candidate for antifungal medication, especially in conditions involving *Candida* spp.

## Figures and Tables

**Figure 1 biotech-13-00016-f001:**
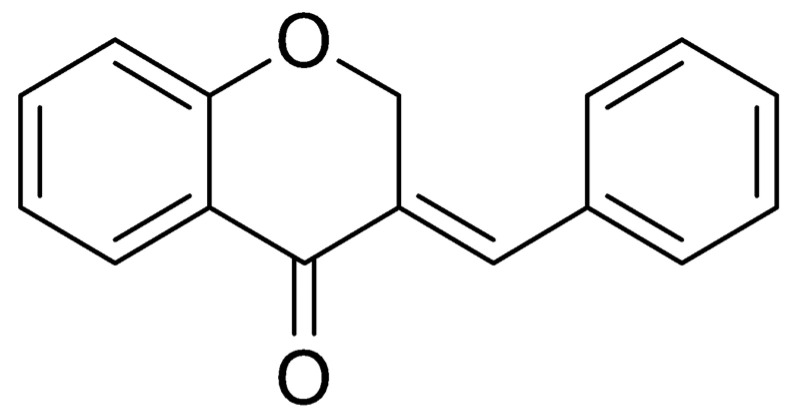
(E)-benzylidene-chroman-4-one (AR25).

**Figure 2 biotech-13-00016-f002:**
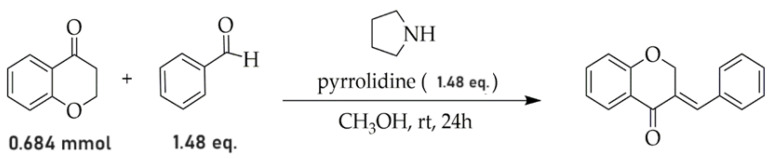
Reaction of 4-chromanone with benzaldehyde in the presence of pyrrolidine.

**Figure 3 biotech-13-00016-f003:**
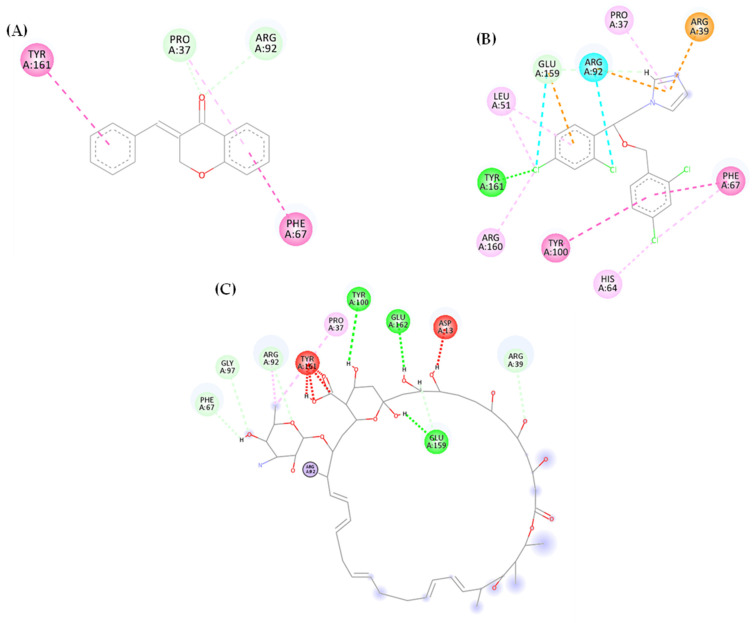
Molecular docking of (E)-3-benzylidene-chroman-4-one (AR25) (**A**), Miconazole (**B**), and Nystatin (**C**) with the molecular target Thymidylate Synthase (5UIV). Pink: alkyl, pi–alkyl, pi–pi T shaped, pi–pi stacked interactions. Lilac: covalent bond. Green: carbon–hydrogen bond and conventional hydrogen bond interactions. Orange: pi–cation and pi–anion interactions. Blue: halogen (Cl, Br, and I) interactions. Red: unfavorable bump interactions.

**Figure 4 biotech-13-00016-f004:**
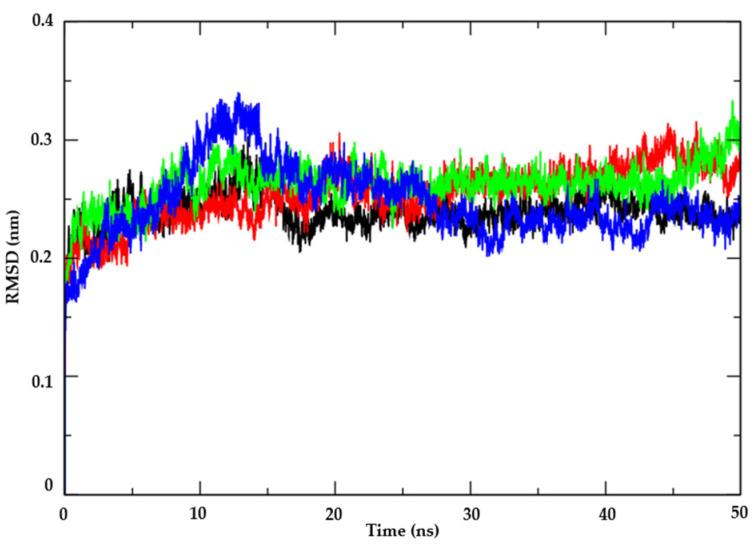
RMSD of the Cα atoms of the enzyme Thymidylate kinase from *Candida albicans* (PDB: 5UIV) (black) and complexed with the compounds AR25 (red), Miconazole (green), and Nystatin (blue).

**Figure 5 biotech-13-00016-f005:**
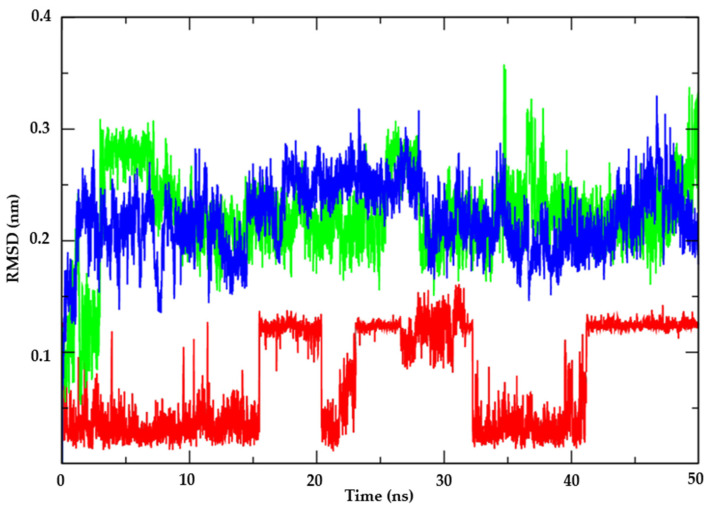
RMSD of the Cα atoms of the compounds. AR25 (red line), Miconazole (green line), and Nystatin (blue line).

**Figure 6 biotech-13-00016-f006:**
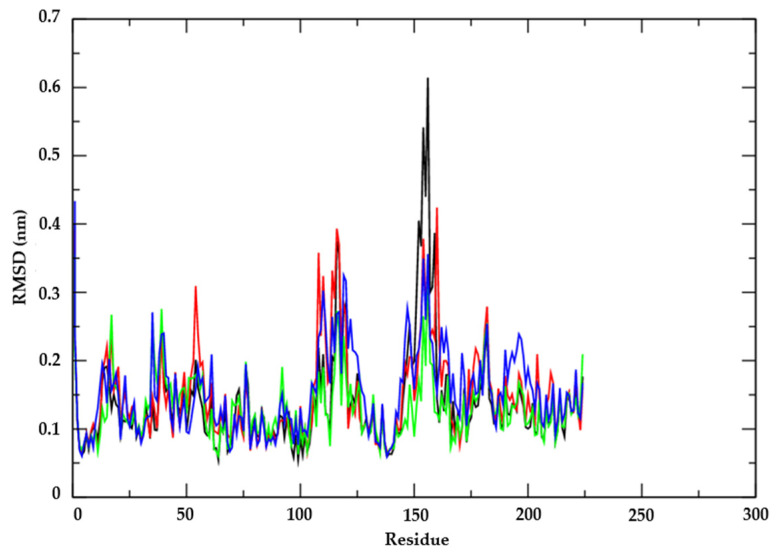
RMSF of Cα atoms. From the enzyme Thymidylate kinase from Candida albicans (PDB: 5UIV) (black line), complexed with the compounds AR25 (red line), Miconazole (green line), and Nystatin (blue line).

**Figure 7 biotech-13-00016-f007:**
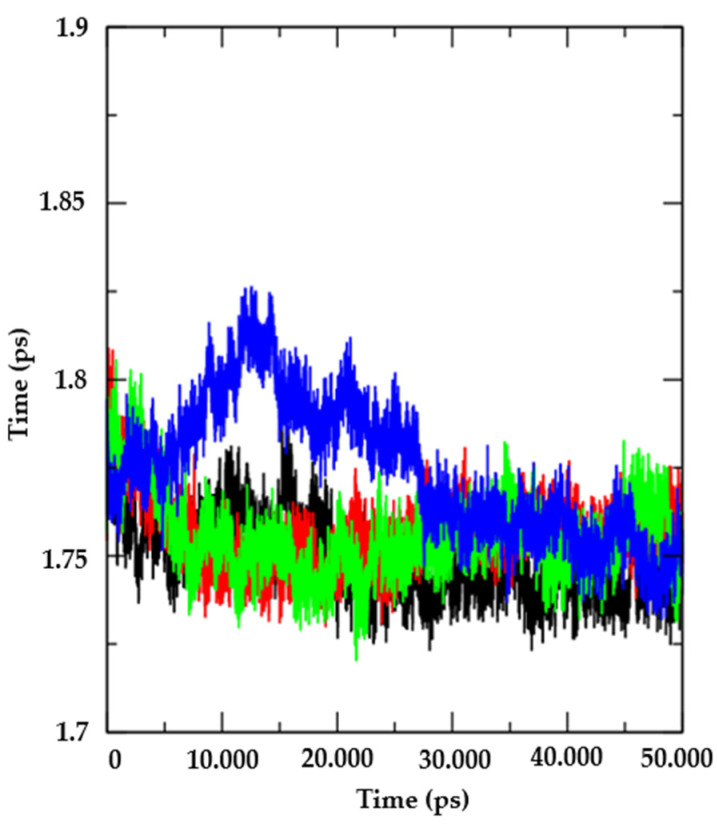
Radius of rotation (Rg). From the Candida albicanns Thymidylate kinase protein (PDB: 5UIV) (black) and complexed with the compounds AR25 (red), Miconazole (green), and Nystatin (blue line).

**Figure 8 biotech-13-00016-f008:**
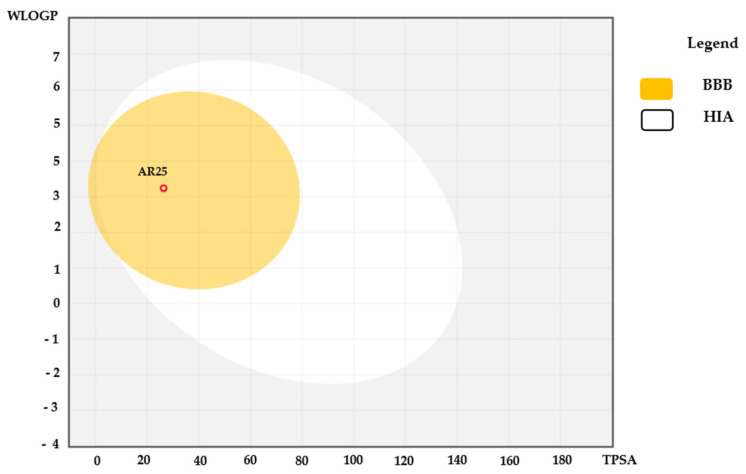
The Boiled Egg diagram illustrates the characteristics of compound AR25. Points located within the yellow area signify molecules predicted to passively permeate through the blood–brain barrier (BBB), while those in the white area denote molecules anticipated to be passively absorbed by the human gastrointestinal tract (HIA). Additionally, red points indicate molecules predicted to be effluxed from the central nervous system by P-glycoprotein.

**Figure 9 biotech-13-00016-f009:**
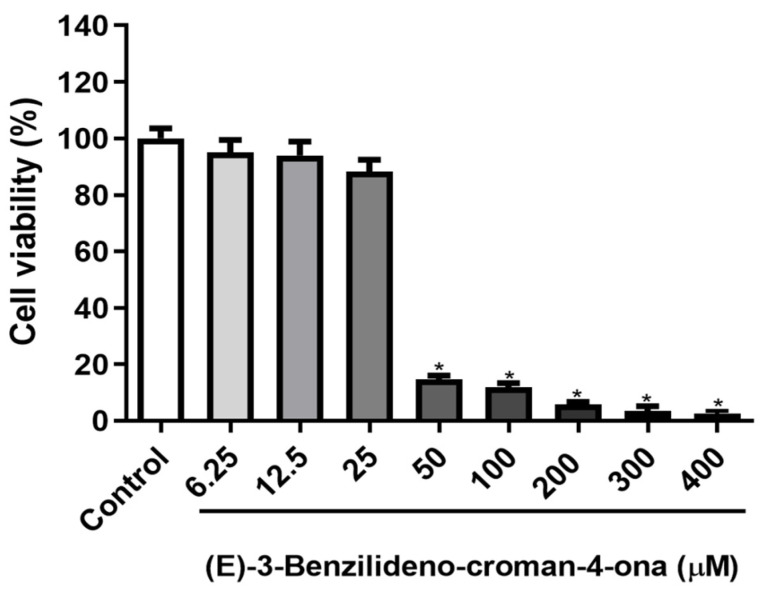
Cytotoxicity of (E)-benzylidene-chroman-4-one on human keratinocytes (HaCaT cell line). Results are expressed as mean ± SD (one-way ANOVA with Tukey’s post-test, * *p* ≤ 0.001).

**Table 1 biotech-13-00016-t001:** Binding energy values (in kJ/mol) of the compounds (E)-3-benzylidene-chroman-4-one (AR25), Nystatin, and Miconazole with molecular targets including 1,3β-glucan synthase (1EQP), squalene epoxidase (4MAI), δ-14-sterol reductase (4QUV), 14-α-demethylase (5TZ1), and thymidylate synthase (5UIV).

Substances	Molecular Targets				
	1EQP	4MAI	4QUV	5TZ1	5UIV
(E)-benzylidene-chroman-4-one	−66.4624	−62.0636	−67.2416	−56.1563	**−102.589**
Miconazole	−106.367	−91.5091	−98.4458	−82.8594	−133.473
Nystatin	−174.305	−157.953	−210.923	−75.3106	−91.3965
PDB Ligand	-	-	-	−85.3911	-

**Table 2 biotech-13-00016-t002:** Coulomb and Lennard-Jones interaction energy values.

Energy	Thymidylate Kinase (PDB: 5UIV)		
	AR25	Miconazole	Nystatin
**Coulomb (C)**	**−51.4842**	−48.0814	**−223.375**
**Lennard-Jones (LJ)**	−97.2097	**−188.998**	−182.996

The lowest energy is presented in bold.

**Table 3 biotech-13-00016-t003:** Key results of ADMET predictions for the compound in the study: gastrointestinal absorption (GI), blood–brain barrier permeation (BHE), violation of the Lipinski rule, mutagenicity, tumorigenicity, reproductive effect, and skin irritancy.

ID	GI ^1^	BBB ^2^	Lipinski Violations	Mutagenicity	Tumorigenicity	Reproductive Effect	Irritant Skin Effect
AR25	High	Yes	0	None	None	None	None

^1^ Gastrointestinal absorption (GI) and ^2^ blood–brain barrier permeation (BBB).

**Table 4 biotech-13-00016-t004:** Predicted interaction with cytochromes P450 (CYP) for the seven compounds (AR25).

ID	CYP				
	CYP1A2	CYP2C19	CYP2C9	CYP2D6	CYP3A4
AR25	Yes	Yes	No	No	No

**Table 5 biotech-13-00016-t005:** Minimum inhibitory concentration (MIC) and minimum fungicidal concentration (MFC) of the molecule (E)-3-benzylidene-chroman-4-one and Nystatin against *Candida* spp. MIC and MFC values are expressed in μg/mL (μM).

	(E)-3-Benzylidene-Chroman-4-One			Nystatin		
Strains	MIC	MFC	MFC/ MIC	MIC	MFC	MFC/ MIC
*C. albicans* ATCC 10231	62.5 (264.52)	62.5 (264.52)	1	3 (3.23)	3 (3.23)	1
*C. krusei* ATCC 6258	62.5 (264.52)	62.5 (264.52)	1	3 (3.23)	3 (3.23)	1
*C. glabrata* ATCC 90030	250.0 (1058.11)	250.0 (1058.11)	1	1.5 (1.61)	1.5 (1.61)	1
*C. tropicalis* ATCC 750	250.0 (1058.11)	250.0 (1058.11)	1	1.5 (1.61)	1.5 (1.61)	1
*C. parapsilosis* ATCC 22019	500.0 (2116.22)	1000.0 (423.44)	2	1.5 (1.61)	1.5 (1.61)	1
*C. albicans* ATCC 60193	1000.0 (4232.44)	1000.0 (4232.44)	1	1.5 (1.61)	1.5 (1.61)	1
*C. albicans* ATCC 90028	500.0 (2116.22)	500.0 (2116.22)	1	1.5 (1.61)	1.5 (1.61)	1

**Table 6 biotech-13-00016-t006:** MIC values of (E)-benzylidene-chroman-4-one and Nystatin in the absence and presence of exogenous ergosterol against *C. albicans* ATCC 90028 and *C. krusei* ATCC 6258 strains. Values expressed in μg/mL (μM).

	(E)-3-Benzylidene-Chroman-4-One					Nystatin			
Concentration (μg/mL)	*C. albicans* ATCC 90028		*C. krusei* ATCC 6258		Concentration (μg/mL)	*C. albicans* ATCC 90028		*C. krusei* ATCC 6258	
	Absence of Ergosterol	Presence of Ergosterol	Absence of Ergosterol	Presence of Ergosterol		Absence of Ergosterol	Presence of Ergosterol	Absence of Ergosterol	Presence of Ergosterol
1000	−	−	−	−	48	−	−	−	−
500	−	+	−	+	24	−	+	−	+
250	+	+	−	+	12	−	+	−	+
125	+	+	−	+	6	−	+	−	+
62.5	+	+	−	+	3	−	+	−	+
31.25	+	+	+	+	1.5	−	+	+	+
15.62	+	+	+	+	0.75	+	+	+	+
7.81	+	+	+	+	0.37	+	+	+	+

Notes: +, fungal growth; −, no fungal growth.

**Table 7 biotech-13-00016-t007:** Values of MIC of the molecule (E)-benzylidene-chroman-4-one and Caspofungin^®^ in the absence and presence of sorbitol (0.8 M) against strains of *C. albicans* ATCC 90028 and *C. krusei* ATCC 6258. Values expressed in μg/mL.

	(E)-3-Benzylidene-Chroman-4-One					Caspofungin^®^			
Concentration (μg/mL)	*C. albicans* ATCC 90028		*C. krusei* ATCC 6258		Concentration (μg/mL)	*C. albicans* ATCC 90028		*C. krusei* ATCC 6258	
	Absence of Ergosterol	Presence of Ergosterol	Absence of Ergosterol	Presence of Ergosterol		Absence of Ergosterol	Presence of Ergosterol	Absence of Ergosterol	Presence of Ergosterol
1000	−	−	−	−	4	−	−	−	−
500	−	−	−	−	2	−	+	−	−
250	+	+	−	−	1	−	+	−	+
125	+	+	−	−	0.5	−	+	−	+
62.5	+	+	−	−	0.25	−	+	−	+
31.25	+	+	+	+	0.125	−	+	+	+
15.62	+	+	+	+	0.062	+	+	+	+
7.81	+	+	+	+	0.031	+	+	+	+

Notes: +, fungal growth; −, no fungal growth.

## Data Availability

Data can be requested by contacting the corresponding author.

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
