# Peer review of "In Silico and In Vitro Evaluation of the Antifungal Activity of a New Chromone Derivative against Candida spp."

_biotech, 2024, doi:10.3390/biotech13020016_

Round 1

Reviewer 1 Report

Comments and Suggestions for Authors

overall, the paper provides valuable insights into the antifungal activity  (E)-benzylidene-chroman-4-one against Candida spp. The inclusion of both in silico and laboratory tests enhances the credability of the findings. However, there are some areas where the paper could be improved:

1. The results of the study are very limited. The authors should add more relevant parameters to enhance the quality of the paper.

2. There should be more detailed description of the experimental procedures employed in the silico and laboratory tests.

3. In depth discussion is needed to explore the interaction between the compound and the fungal membrane components.

4. was there any in vivo study associated with this paper? if yes, why it has not been added into this paper. Currently, the study is very limited.

Author Response

# Reviewer 1

Overall, the paper provides valuable insights into the antifungal activity  (E)-benzylidene-chroman-4-one against Candida spp. The inclusion of both in silico and laboratory tests enhances the credability of the findings. However, there are some areas where the paper could be improved:

  1. The results of the study are very limited. The authors should add more relevant parameters to enhance the quality of the paper.

Response: New tests, such as molecular dynamics, were added to the ADMET predictions article in order to improve the quality of the article.

  1. There should be more detailed description of the experimental procedures employed in the silico and laboratory tests.

Response: As suggested, adjustments were made to the description of the experimental procedures, better detailing the methodology of each test.

  1. In depth discussion is needed to explore the interaction between the compound and the fungal membrane components.

Response: As suggested, the interaction between the compound and components of the fungal membrane was further studied, however, for a greater understanding of the mechanism of action of the compound, further studies are needed to evaluate this parameter, as discussed in the text.

  1. was there any in vivo study associated with this paper? if yes, why it has not been added into this paper. Currently, the study is very limited.

Response: The initial purpose of the study was to provide insights into the antifungal potential of the compound through in vitro and in silico methodologies, aiming to elucidate possible pharmacological mechanisms. These study strategies aim to make the discovery process more accurate and reduce the use of animals in experimentation. After the in vitro and in silico validation, the second part of the research will involve pre-clinical and clinical stages in dental conditions caused by Candida albicans.

Reviewer 2 Report

Comments and Suggestions for Authors

1.       Line 27 appears to be a heading within a paragraph. Please check and ensure proper formatting. 

2.       Inconsistencies were found in the numbering of headings and subheadings throughout the manuscript. Please review and ensure consistent formatting.

3.       Specify the source of the Candida strains used in the study (e.g., clinical isolates, culture collections).

4.       Explain the rationale behind choosing three Candida albicans strains and only one strain for each other species.

5.       Mention the specific version of the CLSI method used for the MIC assay. Provide a reference for the chosen method.

6.       Justify using only keratinocyte cells (HaCaT cells) to evaluate cytotoxicity. Are there other relevant cell lines that could be considered for a more comprehensive evaluation?

7.       Expand on the need for further cytotoxicity analyses. Briefly elaborate on the type of analyses recommended (e.g., evaluation on different cell lines).

8.       Table 2 lacks statistical analysis. Include data analysis for all results.

9.       Explain the rationale behind choosing only C. albicans ATCC 90028 and C. krusei ATCC 6258 for the ergosterol and sorbitol assays (Tables 2 & 3). Why weren't these assays performed on the other Candida strains used in the study?

10.   Figure 3 is mentioned twice. Check all figure and table numbers again to avoid errors.

11.   Improve clarity in Figure 3. Provide the exact p-values for all results, including non-significant ones, instead of just "p ≤ 0.001".

12.   Improve the conclusion section by summarizing the key findings and their broader implications.

13.   Discuss potential future research directions to build upon these findings.

14.   Clearly outline the limitations of the current study.

15.   Briefly mention the limitations of in silico docking analysis and the need for in vitro validation (as presented in this study).

16.   When comparing the activity of (E)-benzylidene-chroman-4-one to other antifungal agents, mention the class to which they belong (e.g., polyene for Nystatin).

17.   Discuss the selectivity of the compound, mentioning whether it has activity against other fungal species beyond Candida.

18.   Consider adding a sentence about the limitations of current antifungal therapies (e.g., emergence of drug resistance) to the introduction section.

Author Response

# Reviewer 2

  1. Line 27 appears to be a heading within a paragraph. Please check and ensure proper formatting. 

R: Corrected as suggested.

  1. Inconsistencies were found in the numbering of headings and subheadings throughout the manuscript. Please review and ensure consistent formatting.

R: Corrected as suggested.

  1. Specify the source of the Candida strains used in the study (e.g., clinical isolates, culture collections).

R: The strains used were reference strains of Candida species obtained from the American Type Culture Collection (ATCC, Rockville, MD, USA).

  1. Explain the rationale behind choosing three Candida albicans strains and only one strain for each other species.

R: The reason behind the choice lies in the fact that the highest number of fungal infections observed in the clinic are caused by the species C. albicans. Thus, the study of chromone against different strains of the species in question was carried out.

  1. Mention the specific version of the CLSI method used for the MIC assay. Provide a reference for the chosen method.

R: CLSI, 2008. Reference Method for Broth Dilution Antifungal Susceptibility Testing of Yeasts: Approved Standard – Third Edition. CLSI Document M27-A3. Clinical and Laboratory Standards Institute, Wayne, PA.

  1. Justify using only keratinocyte cells (HaCaT cells) to evaluate cytotoxicity. Are there other relevant cell lines that could be considered for a more comprehensive evaluation?

R: One of the possible pharmacological applications of the compound under study will be for topical applications – oral or genital conditions caused by Candida albicans. For this reason, the use of human keratinocyte cell lines (HaCat) was standardized to determine the cytotoxic potential of the drug tested by evaluating its potential for cell damage in the dermis.

  1. Expand on the need for further cytotoxicity analyses. Briefly elaborate on the type of analyses recommended (e.g., evaluation on different cell lines).

R: Added as suggested

  1. Table 2 lacks statistical analysis. Include data analysis for all results.

R: Following the protocol indicated in CLSI 2008, results for MIC and MBC are determined using the mode, through descriptive analysis; thus, statistical analysis cannot be applied to these results.

  1. Explain the rationale behind choosing only C. albicans ATCC 90028 and C. krusei ATCC 6258 for the ergosterol and sorbitol assays (Tables 2 & 3). Why weren't these assays performed on the other Candida strains used in the study?
  2. Figure 3 is mentioned twice. Check all figure and table numbers again to avoid errors.

R: Corrected as suggested

  1. Improve clarity in Figure 3. Provide the exact p-values for all results, including non-significant ones, instead of just "p ≤ 0.001".

R: The exact values were not provided because they were not expressed in tables. The authors chose to present the results in the form of column graphs, using an asterisk to indicate the significance of the results compared to the control group.

  1. Improve the conclusion section by summarizing the key findings and their broader implications.

R: As suggested, the conclusions section was improved.

  1. Discuss potential future research directions to build upon these findings.

R: The future directions of the research will involve new tests for better assessment of cytotoxicity, new tests for a better understanding of the molecule's mechanism of action, and then moving towards pre-clinical and clinical stages with the aim of developing a formulation applicable to this compound.

  1. Clearly outline the limitations of the current study.

R: Need for further studies to evaluate other toxicity parameters; additional mechanism of action tests for a better understanding of the compound's action; lack of in vivo experiments; absence of tests with other fungal genera.

  1. Briefly mention the limitations of in silico docking analysis and the need for in vitro validation (as presented in this study).

R: In silico techniques are applied preliminarily to guide the next steps of the research. Based on the targets studied in docking, it is possible to perform a comparison using in vitro tests, complementing and corroborating what was obtained in docking, thereby increasing the reliability of the results.

  1. When comparing the activity of (E)-benzylidene-chroman-4-one to other antifungal agents, mention the class to which they belong (e.g., polyene for Nystatin).

R: Corrected as suggested

  1. Discuss the selectivity of the compound, mentioning whether it has activity against other fungal species beyond Candida.

R: The tests used for the present study were conducted only with strains of the Candida genus, so it is not possible to assert activity with other fungal species besides Candida for this compound. It would be necessary to perform new tests to evaluate the antifungal effect of this molecule on other fungal genera. However, according to the literature, chromone, the molecule from which AR25 is derived, exhibits activity against other fungal species besides Candida.

  1. Consider adding a sentence about the limitations of current antifungal therapies (e.g., emergence of drug resistance) to the introduction section.

R: Considering the suggestion, a sentence was added to the introduction about the limitations of current antifungal therapies.

Reviewer 3 Report

Comments and Suggestions for Authors

The manuscript titled "In Silico and In Vitro Evaluation of the Antifungal Activity of a New Chromone Derivative against Candida spp." authored by Gleycyelly Rodrigues Araújo et al. provides a thorough analysis of the antifungal properties of chromone derivatives, using both computational and empirical methods. The authors utilize computational techniques to elucidate how the novel compound (E)-benzylidene-chroman-4-one targets and disrupts the cell membranes of Candida species, which are noteworthy due to their widespread prevalence and resistance to current antifungal therapies. It is important to experimentally verify the predicted mechanisms of action. If the authors lack the necessary infrastructure or expertise to conduct these experiments, they should acknowledge this limitation in the discussion section. I recommend that the manuscript be accepted for publication, subject to the completion of revisions in line with the comments provided below. This would ensure that the final publication reflects a thorough evaluation and discussion of the novel compound's potential, enhancing its scientific rigor and relevance to the field of antifungal drug development.

Comments to the Authors:

  1. While the manuscript effectively discusses the antifungal properties of the new chromone derivative, a direct comparative analysis of this compound against current antifungal treatments would be highly beneficial. Such a study should evaluate not only the efficacy but also the safety and cost implications of the new compound. This comprehensive comparison would help clarify the potential advantages or disadvantages of the new compound relative to existing treatments, thereby better positioning it within the current therapeutic landscape.
  2. It would be valuable to elaborate further in the discussion section to provide a more rounded critique and potentially guide future research directions. This enhancement would increase the manuscript's contribution to the field of antifungal drug development. A deeper analysis could include suggestions for addressing the limitations identified in the current study, exploring the compound's mechanism of action in more detail, or proposing additional studies to assess the long-term efficacy and safety of the compound in clinical settings. Additionally, discussing the implications of the research findings in the context of the ongoing global challenge of antifungal resistance would also enrich the manuscript.
  3. Methods section, page 3, lines 105-119: Was each molecule represented by a single conformation in the SDF file? Additionally, do the authors believe that the use of semi-empirical methods in the minimization process provides added value to the in-silico study?

Author Response

# Reviewer 3

The manuscript titled "In Silico and In Vitro Evaluation of the Antifungal Activity of a New Chromone Derivative against Candida spp." authored by Gleycyelly Rodrigues Araújo et al. provides a thorough analysis of the antifungal properties of chromone derivatives, using both computational and empirical methods. The authors utilize computational techniques to elucidate how the novel compound (E)-benzylidene-chroman-4-one targets and disrupts the cell membranes of Candida species, which are noteworthy due to their widespread prevalence and resistance to current antifungal therapies. It is important to experimentally verify the predicted mechanisms of action. If the authors lack the necessary infrastructure or expertise to conduct these experiments, they should acknowledge this limitation in the discussion section. I recommend that the manuscript be accepted for publication, subject to the completion of revisions in line with the comments provided below. This would ensure that the final publication reflects a thorough evaluation and discussion of the novel compound's potential, enhancing its scientific rigor and relevance to the field of antifungal drug development.

Comments to the Authors:

  1. While the manuscript effectively discusses the antifungal properties of the new chromone derivative, a direct comparative analysis of this compound against current antifungal treatments would be highly beneficial. Such a study should evaluate not only the efficacy but also the safety and cost implications of the new compound. This comprehensive comparison would help clarify the potential advantages or disadvantages of the new compound relative to existing treatments, thereby better positioning it within the current therapeutic landscape.
  2. It would be valuable to elaborate further in the discussion section to provide a more rounded critique and potentially guide future research directions. This enhancement would increase the manuscript's contribution to the field of antifungal drug development. A deeper analysis could include suggestions for addressing the limitations identified in the current study, exploring the compound's mechanism of action in more detail, or proposing additional studies to assess the long-term efficacy and safety of the compound in clinical settings. Additionally, discussing the implications of the research findings in the context of the ongoing global challenge of antifungal resistance would also enrich the manuscript.
  3. Methods section, page 3, lines 105-119: Was each molecule represented by a single conformation in the SDF file? Additionally, do the authors believe that the use of semi-empirical methods in the minimization process provides added value to the in-silico study?

R: We agree with your suggestion. We have included in the discussion information about limitations of the study and perspectives for conducting new investigations that allow for the expansion of knowledge on the mechanisms of antifungal activity and evaluation of this effect in in vivo models, including aspects related to the toxicity and safety of the compound.

Reviewer 4 Report

Comments and Suggestions for Authors

Luciana Scott and his coworkers have reported: “In Silico and In Vitro Evaluation of the Antifungal Activity of a new Chromone Derivative against Candida spp.” They synthesized AR25 and did some studies like in silico analysis, In Vitro Analysis, Mechanism of Action on the Fungal Membrane and Cell Wall, and Cell Viability Assay MTT (Cytotoxicity). This manuscript needs some minor corrections mentioned below before publishing in the BioTech journal.

1.     Line 77, It is more compatible with 'structure' than with 'synthetic reaction'.

2.     In Line 82, the authors mention it is a new molecule. However, there have been several reports already that reported this molecule. The authors better remove the word new.

3.     Authors have already reported this molecule and its antifungal activity in their previous paper. Ref. is DOI: 10.3390/ph15060712. The authors should justify this.

4.     Line 88, the same equivalents should be the same m.moles. The authors should recheck the m.moles of pyrrolidine.

5.     Authors should also do the molecular dynamics (MD) simulation study to find out extensive molecular interactions with the target.

6.     Authors should also report Insilco Ki along with docking energy.

7.     In silicon ADMET analysis should also be performed for the synthesized compound.

Comments on the Quality of English Language

Minor editing of English language required

Author Response

# Reviewer 4

Luciana Scott and his coworkers have reported: “In Silico and In Vitro Evaluation of the Antifungal Activity of a new Chromone Derivative against Candida spp.” They synthesized AR25 and did some studies like in silico analysis, In Vitro Analysis, Mechanism of Action on the Fungal Membrane and Cell Wall, and Cell Viability Assay MTT (Cytotoxicity). This manuscript needs some minor corrections mentioned below before publishing in the BioTech journal.

  1. Line 77, It is more compatible with 'structure' than with 'synthetic reaction'.

R: Corrected as suggested

  1. In Line 82, the authors mention it is a new molecule. However, there have been several reports already that reported this molecule. The authors better remove the word new.

R: Corrected as suggested

  1. Authors have already reported this molecule and its antifungal activity in their previous paper. Ref. is DOI: 10.3390/ph15060712. The authors should justify this.

R: The molecule and its antifungal activity were reported in the cited work as it is the article related to the molecule's synthesis, which describes its acquisition process and presents initial tests only to confirm if the compound exhibits antifungal potential. However, our study is more in-depth and seeks to confirm the antifungal potential of the compound and assess it in other Candida species, as well as to understand its mechanism of action and obtain responses regarding the compound's affinity to fungal targets, tests that are conducted with the aim of making the discovery process more precise and reducing the use of animals in experimentation. With in vitro and in silico validation, the research can advance to pre-clinical and clinical stages in dental conditions caused by Candida albicans. Thus, the justification has also been added to the manuscript.

  1. Line 88, the same equivalents should be the same m.moles. The authors should recheck the m.moles of pyrrolidine.

R: Corrected as suggested

  1. Authors should also do the molecular dynamics (MD) simulation study to find out extensive molecular interactions with the target.

R: Molecular dynamics simulations with 50ns were added.

  1. Authors should also report Insilco Ki along with docking energy.

R: The Docking Score calculations were carried out in the Molegro Virtual Docker (MVD) software, so in this software the binding energy value is provided in the unit Kj.mol-1 and the program itself does not provide the automatically calculated ki values. Furthermore, when converting to the ki value, the value becomes very small in the order of picomolar (pM), so for this reason they were not added.

  1. In silicon ADMET analysis should also be performed for the synthesized compound.

R: On silicon, ADMET analysis was added as suggested.

Round 2

Reviewer 3 Report

Comments and Suggestions for Authors

I endorse acceptance of the manuscript for publication